Designing a novel technique for evaluation of tourism informatization in scenic spots from a big data perspective

Fu Li 1 2
Yi Yao 1 2 cjxagjgl@163.com
Liu Lina 1 2
Chen Ran 3
1 Department of Management, Taiyuan Normal University , Jinzhong City , China
2 Tourism Information Technology Innovation Center , Jinzhong City, Shanxi Province , China
3 Department of Tourism Management, Binzhou University , Binzhou City , China
Alatas Bilal
Electronic publication date: 2025 Apr 18
Publication date: 2025
Volume: 11
Electronic Location ID: e2807
Received 2024 Aug 15; Accepted 2025 Mar 16
Copyright: © 2025 Fu et al.
Copyright year: 2025
Copyright holder: Fu et al.
License: This is an open access article distributed under the terms of the Creative Commons Attribution License, which permits unrestricted use, distribution, reproduction and adaptation in any medium and for any purpose provided that it is properly attributed. For attribution, the original author(s), title, publication source (PeerJ Computer Science) and either DOI or URL of the article must be cited.
License URL: https://creativecommons.org/licenses/by/4.0/

Keywords: Big data, Data security, Machine learning, Neural networks, Tourist sentiment analysis, BERT, Tourism informatization

Funding: Research on active relationship repair strategy of tourism enterprises in resource-based areas 2021-147 Shanxi Scholarship Council of China The Education Department of Shanxi Province, China The research was founded within the project No. 2021-147 entitled: “Research on active relationship repair strategy of tourism enterprises in resource-based areas”, being a part of Strategic Research Program “Research Project Supported by Shanxi Scholarship Council of China” supported by The Education Department of Shanxi Province, China. The funders had no role in study design, data collection and analysis, decision to publish, or preparation of the manuscript.

==============================
In recent years, tourism has become a significant driver of many countries’ economies. To maximize revenue from tourism, it is crucial to prioritize the effective management of scenic spots and tourist attractions, and also raise awareness about these places. Social media platforms have played a pivotal role in promoting tourism, as users frequently share videos and reviews related to tourism. Analyzing and managing these reviews is essential for understanding tourists’ opinions about specific destinations. In this study, we evaluated a scenic spot by analyzing tourists’ sentiments. Data was collected from popular social media sites such as TripAdvisor and Twitter using web scraping and the Twitter API. The raw data was preprocessed to remove irrelevant information and redundancies and was properly annotated for further processing. We applied two approaches to analyze the sentiments of tourists. First, we vectorized the text representing the sentiment using the term frequency-inverse document frequency (TF-IDF) and utilized big data analytics to extract meaningful insights. Secondly, we employed a pre-trained large language model, bidirectional encoder representations from transformers (BERT), with a linear classifier to classify tourists’ sentiments. The results of the big data analytics approaches were compared with those of BERT and previously proposed methods. BERT outperformed other machine learning models, achieving an average accuracy of 83.5% on the test set. These insights are valuable for evaluating the informatization of tourist spots, destination management, hospitality, and overall tourist attractions.

Introduction

The global tourism industry has seen a significant boost due to the rapid expansion of the global economy, emerging as a major source of income worldwide (Li, Deng & Cai, 2020). With increasing living standards, people have become more knowledgeable about tourism, leading to a surge in the number of visitors to beautiful destinations around the world. Despite significant growth in China’s tourism sector, there remains a substantial gap in digitization compared to industrialized nations due to various challenges (Khurramov, 2020). The majority of tourist destinations do not have an overarching tourism informatization development plan, and there is not enough money invested in building this infrastructure. While some information technology is present in tourist destinations, it is restricted to low-end computers for information management (Xu et al., 2024). In contrast, there are limited tourist information consultation centers, multimedia information query systems, automatic voice interpretation systems, and other supporting resources. Consequently, the management of scenic places is unable to understand the operation conditions of scenic spots in real-time, and therefore, cannot understand the sentiment and intention of the tourist about the scenic spot (Xu et al., 2024). Thankfully, with the recent advances in information and communication technology (ICT) such as deep learning and big data technology (Pencarelli, 2020), faster and more efficient data collection and processing become feasible.

By employing modern ICT, tourists have enhanced their negotiating power and acquired access to information as they have grown more skilled and digitally educated (Lee et al., 2021). As people become more digitally proficient, the internet, social media, and big data have significantly transformed the way individuals travel around the globe (Bano, Liu & Khan, 2022). Before their trips, individuals often plan their trips by utilizing online resources to select their destinations to optimize their travel experiences and minimize resource consumption. In subsequent phases such as actual travel, and post-travel, social media platforms are frequently used by individuals to share their travel experiences and memories (Leelawat et al., 2022). Consequently, social media has emerged as the most widely used platform for opinions and inspiration among Internet users looking for new travel destinations, things to do while visiting, and leisure and entertainment options. A massive amount of data and comments are generated and routinely shared due to the growth of travel-related websites and social networks. Travelers use these statistics, along with reviews from other travelers, as valuable information sources when choosing destinations and areas of interest for their trips (Mariani & Baggio, 2022). This data is large in volume, generated at a fast speed, and has a diverse nature therefore considered as big data.

Big data has created new opportunities for better decision support and might be seen as a “new era” of the data-driven paradigm (Sarkar et al., 2022). Despite the preliminary nature of the research (primarily for information systems design), these ideas have immediate application in data-intensive industries like tourism. Examples in the tourist industry include destination marketing, hospitality management, customer relationship management, and knowledge development for strategic planning objectives in tourism destinations Social media platforms serve as vast archives of user-generated content, offering real time insights into the preferences, behaviors, and feelings of tourists. However, the analysis of this data presents difficulties owing to its unstructured nature, swift growth, and diverse types, which require sophisticated tools and techniques for effective evaluation.

Although there is a wealth of user-generated content, most studies currently concentrate on individual applications of big data and sentiment analysis in tourism. They frequently neglect the broader implications for real-time data informatization and decision-making at tourist sites. The analysis of social media data is made complex due to its unstructured and ever-changing nature, resulting in a significant gap in research that thoroughly addresses these issues. Additionally, even though contemporary information and communication technology (ICT) tools, like big data platforms and deep learning models, show great potential, their integration into practical frameworks aimed at enhancing tourist satisfaction and management processes is still largely unexamined (Hu et al., 2024).

This study addresses these challenges by utilizing bidirectional encoder representations from transformers (BERT) for sentiment analysis alongside Hadoop for efficient big data processing. By integrating these technologies, we deliver a robust framework for the real-time assessment of tourist sentiments, providing actionable insights for managing scenic spots and improving overall informatization. While some earlier research has applied BERT for sentiment analysis in tourism, our study makes significant advances in this area in several important ways. Firstly, unlikely existing research that relies solely on BERT, we combine big data technologies like Hadoop to effectively handle and analyze diverse, large-scale datasets obtained in real-time from Twitter and TripAdvisor. Secondly, our research performs a thorough comparison between BERT and several traditional machine learning models, such as multinomial naïve Bayes (MNB), k-nearest neighbors (KNN), and random forest (RF), establishing a solid evaluation framework that underscores BERT’s advantages. Finally, our article focuses on the informatization of scenic spots, utilizing tourist sentiment as a metric for real-time evaluation and management, which is an aspect not explicitly explored in prior BERT-oriented studies. We consider two famous social media networks for the data collection. The first one is Twitter and the second one is TripAdvisor. The Twitter data is collected through the Twitter API, and the textual data from TripAdvisor is collected through scraping. The collected data represents the sentiment of tourists about a specific scenic spot which is then preprocessed and annotated accordingly. After data preprocessing, the big data analytics and BERT model are employed to categorize the sentiment of tourists. The findings from this research can assist tourism managers in pinpointing improvement areas, optimizing resource distribution, and boosting tourist satisfaction. Additionally, the methodology is applicable to other industries that depend significantly on sentiment-driven decisions. Furthermore, it can support tourism management in making enhancements and offering improved hospitality to tourists. The basic contributions of the article are: This article targets famous social media sites, Twitter and TripAdvisor, for data collection and shows the data collection process in real-time for a particular scenic spot using mobile-embedded sensors. A Twitter API is used to collect data from Twitter while scraping is used to mine data from TripAdvisor. The collected is in the raw form which is then cleaned, annotated, and preprocessed. To manage the variety and speed of data from social media and real-time IoT sources, the study employs Hadoop-based big data management and storage solutions. This framework facilitates effective processing, storage, and handling of large, diverse datasets, guaranteeing scalability and adaptability to meet the rapidly expanding data needs in tourism informatization.

This article evaluates the informatization of scenic spots by considering reviews posted on social media that show the sentiment of tourists about a tourist location. The sentiments of tourists are classified by using different ML, and DL techniques including MNB, KNN, RF, voting classifier (VC), XGBoost classifier (XGB), staking classifier (SC), and BERT with linear classifier. Before applying ML algorithms, the text is vectorized or embedded through the term frequency-inverse document frequency (TF-IDF) method, which before applying the linear classifier the BERT model is used to compute the contextual aware embedding vector.

In this article, a comprehensive evaluation of the performance of all the machine learning ML and DL techniques utilized in the article is conducted. The results are thoroughly compared with each other and with other models proposed in the existing literature. Firstly, a detailed comparison among the proposed algorithms is made in terms of accuracy, precision, recall, and F1-score. Secondly, the performance of the top model, which is BERT with a linear classifier, is compared with the existing works in the field.

The study links sentiment analysis results to actionable insights for managing scenic spots, showcasing practical applications of its findings. Stakeholders can use these results to pinpoint improvement areas, boost visitor satisfaction, and propel the digital transformation of tourism services.

In contrast to broader studies on tourism sentiment analysis, this article specifically focuses on major scenic locations in China, providing a localized and specialized assessment. The proposed approach guarantees that the results can be directly utilized to enhance tourism management and informatization at key destinations.

The rest of the article is structured in the following sections. In the “Literature Review”, all the similar works done so far in the literature are summarized by discussing their pros and cons. In the “Evaluation of Tourism Informatization using Big Data”, the evaluation of tourism informatization using big data is carried out. “Evaluating Scenic Spot through Tourist Sentiment Analysis Using BERT” evaluates scenic spots through tourist sentiment analysis using BERT with a linear classifier on the top. “Result and Discussion” is about the result and discussion which validate the proposed methodology through results. “Conclusions” finally concludes the overall theme of the article.

Literature review

The tourism sector become one of the main tires of the economy of different countries around the globe, even the economy of some countries solely depends on tourism. Therefore, researchers and enterprises also pay attention to the tourism sector. The idea of tourism network information, the data collected by travelers searching for information about tourism destinations throughout their trip, has grown with the introduction and advancement of the Internet. The challenge of performing a multidimensional analysis of tourism destination management and tourist satisfaction using data from tourism networks merits further research.

Machine learning and deep learning based approaches

A few academics have evaluated tourism destinations by looking at their geographical locations. For example, Zhang, Chen & Li’s (2019) work used computer deep learning technology to categorize 35,356 Flickr photographs taken in Beijing by tourists into 103 settings. They carried out statistical research to compare the actions and attitudes of tourists from various countries and continents. Likewise, Luo et al.’s (2023) study offers decision support for travelers suggesting beautiful spots and matching recommendations for the management of picturesque locations.

By using recommendation-based computing systems, sentiment analysis can help tourists. The software can accomplish this by first collecting the salient features of probable travel destinations (such as hotels), then extracting reviews about these locations, use aspect-based sentiment analysis to process the reviews, and then summarizing the general sentiment expressed about the customers’ possible options (Manosso & Domareski Ruiz, 2021). In Paolanti et al. (2021), presented a method to analyze public opinion on Cilento, a popular tourist destination in Southern Italy. Their approach is based on a recently compiled collection of tweets about travel. To define the spatial, temporal, and demographic tourist movements throughout the wide area of this rural tourism zone and along its coasts, they demonstrated and tested a DL social geodata framework.

A study by Leelawat et al. (2022) suggested, in which they classified the sentiment and intention classes of tweets in English about travel to Phuket, Chiang Mai, and Bangkok. Next, they examined how well three machine learning algorithms—decision tree, random forest, and support vector machine—predicted the intentions and feelings expressed in the tweets. With a maximum accuracy of 77.4%, the support vector machine technique yielded the best results for sentiment analysis. Li, Gao & Song (2023) also used Internet data from several sources to predict the number of visitors to two Chinese tourist destinations. They consider the granular sentiment analysis to analyze the behavior of tourist toward these tourist destinations.

Big data-based approaches

Big data from social media platforms has produced several opportunities for decision-makers by providing additional information and insight, especially in the tourism sector. According to Seraphin (2020), big data and analytics are two of the most important technologies that will affect the travel and hospitality sector over the next five years. They considered the potential and threats that these technological advancements bring to tourist and travel agencies, emphasizing the need of data governance and procedures for morally and practically sound data management in the travel and hospitality industries. The development of analytics and big data (BD) in the travel and hospitality industries is covered by Mariani (2020). It outlines how BD can change going forward and examines the significant role it has played in tourism and hospitality research thus far. Similarly, to create an agenda for future study, Mariani & Baggio (2022) examined the corpus of literature related to big data and analytics in the hospitality and tourism industries. They did this by identifying macro topical themes, research streams, and gaps.

Large language model-based approaches

Viñán-Ludeña & de Campos (2022) explore sentiment analysis methods applied to social media platforms like Twitter, Facebook, Instagram, and TripAdvisor to better understand tourist experiences and preferences. They present a Spanish-Tourism-BERT model designed for sentiment classification, which adeptly processes multilingual data from various social media sources. By examining content generated by tourists, the research offers essential insights into customer satisfaction, destination reputation, and service quality issues. The findings emphasize how advanced natural language processing models can enhance destination management strategies and streamline the analysis of customer feedback on a large scale. Additionally, their model’s multilingual functionality highlights the need for accommodating different languages in tourism informatics to ensure precise analysis and tailored recommendations for stakeholders. This study serves as a crucial reference for implementing sentiment analysis in data-driven decision-making within the tourism industry.

Similarly, the research by Xu et al. (2023) explores the development of tourism knowledge graphs to efficiently manage and integrate large-scale data. By employing advanced techniques such as bidirectional long short-term memory (BiLSTM), BERT, and attention mechanisms, the authors emphasized entity recognition and relationship extraction to transform unstructured tourism data into structured knowledge graphs. This strategy not only reduces data redundancy but also facilitates the integration of varied datasets for thorough analysis. Moreover, the incorporation of emotion-based popularity analysis yields deeper insights into tourist preferences and behaviors. The knowledge graph acts as a foundational framework for smart tourism applications like recommendation systems, trend analysis, and service optimization. By merging structured data with emotional insights, this study paves an innovative path for advancing tourism informatics and fostering data-driven decision-making to enhance the tourist experience. This work establishes a solid technical basis and underlines the significance of incorporating advanced machine learning models into tourism data management systems.

Gaps and proposed methodology

Although existing studies offer valuable insights, they also face several limitations. Traditional machine learning methods, like those utilized by Leelawat et al. (2022), often struggle with large datasets and multilingual contexts, which limits their scalability and relevance in various tourism scenarios. Likewise, while deep learning approaches such as the Spanish-Tourism-BERT model (Viñán-Ludeña & de Campos, 2022) are proficient at managing complex datasets, their application across global destinations frequently neglects localized cultural, linguistic, and behavioral differences. Big data strategies, exemplified by the tourism knowledge graph proposed by Xu et al. (2023), prioritize data integration and reducing redundancy but often ignore the importance of real-time sentiment analysis, which is vital for effective tourism management. Additionally, granular sentiment analysis techniques employed by Li, Gao & Song (2023) tend to concentrate on forecasting visitor numbers and general sentiment patterns, creating shortcomings in the thorough assessment of tourist satisfaction and management of attractions.

To tackle these challenges, our proposed study presents an innovative methodology that merges big data techniques with advanced natural language processing, offering a localized and thorough assessment of tourism sentiments at specific scenic locations in China. While previous research has utilized BERT for sentiment analysis in tourism, our study significantly progresses in this field. Firstly, unlike prior studies that depend solely on BERT, we incorporate big data technologies such as Hadoop to effectively manage and analyze diverse, large-scale datasets sourced in real time from platforms like Twitter and TripAdvisor. This approach allows us to overcome the restrictions posed by static datasets and enables dynamic sentiment analysis. Secondly, our research establishes a holistic evaluation framework by conducting an in-depth comparison of BERT with traditional machine learning models, including MNB, KNN, and RF. This comparison underscores BERT’s enhanced accuracy and efficacy in processing multilingual and unstructured tourism data. Lastly, our study uniquely concentrates on the informatization of scenic spots, employing tourist sentiment as a real-time metric for assessing and managing tourism destinations. This focus, which has been largely overlooked in previous BERT-centered studies, offers actionable insights into tourist preferences and behaviors, improving destination management, attraction strategies, and overall hospitality services.

Evaluation of tourism informatization using big data

This section introduces the concept of big data in evaluating tourist behavior and sentiment toward a specific scenic spot. More specifically, it describes the big data source of tourism such as social media, TripAdvisor, and data collection and preprocessing which covers the data extraction using scraping and data preprocessing.

The tourism sector is one of the main drivers of the economies of many nations (Khalid, Okafor & Burzynska, 2021; Zhao et al., 2024). It also significantly boosts GDP and employment in many nations and areas, particularly the EU, where five of its member states rank among the top ten travel destinations worldwide (Go & Kang, 2023). To attract more tourists to a scenic spot, all the data related to tourists and tourism should be properly managed. The tourism-related data in the present age is heterogeneous as collected from different sources including social media, blogging sites, and different IoT-based devices installed in the scenic spot. Figure 1 shows the three layers architecture for the evaluation of tourism information.

Figure 1 Layer architecture of the evaluation of tourism informatization in scenic spots from a big data perspective.

Big data architecture and framework

The big data architecture proposed for the assessment of tourist informatization is illustrated in Figure. There are three layers in the architecture: Data acquisition layer, data transfer layer, and data processing layer.

The first layer is the data acquisition layer which collects data from scenic spots and tourism using different devices. Some devices such as cameras, sensors, and user smartphones collect real-time data from the scenic spot, which is crucial for monitoring visitor activity. But in this study, we only look at the sensors that are built into cellphones, such accelerometers for activity tracking and GPS for geo-location tracking. Contextual information regarding tourist activities is indirectly provided by these smartphone sensors, such as location-based tagging of images and videos posted on social media. Tourists post images and videos of the visited place on social media and blogging sites, which later receive feedback and reviews from the public. Information from social media and other tourism-related sites is very important to be collected in order to evaluate the sentiment and intention of the public about the scenic spot. We mainly consider three types of online sites for data collection including Twitter, Baidu, and TripAdvisor. Due to the limited storage and processing capacity of the data collection node, the data needs to be transmitted to the server for further processing and valuable insight extraction. The optimal way of data transmission to the remote server is the wireless method because most of the time the beautiful and scenic spots are at the remote location. The local networks, and cellular networks including 3G, 4G, and 5G are considered primary sources of data transmission these days. The transmitted data to the server is heterogeneous and has a large volume therefore, cannot be processed with traditional techniques. As a result, big data analytics is applied to that data to extract meaningful insight.

Social media as a big data source of tourism

According to its definition, social media also known as social networks is a term used to describe a wide range of digital media and technologies that facilitate collaborative action and content creation and sharing among users (Hu et al., 2024). Social media is a powerful source of big data that offers decision-makers highly diverse, fast-moving, and potentially useful data (Mirzaalian & Halpenny, 2021). This big data is produced by individuals across the globe using social networking sites and apps, including Facebook, YouTube, LinkedIn, Twitter, TripAdvisor, TikTok, and Flickr. The content of social networks consists of user-generated blog posts on current events, images, short videos, quick remarks or viewpoints, and private or business-related data. Due to the rapid expansion of social networks (social media), people may now freely share such information with the public, and a vast global society of constantly connected individuals who are eager to share, communicate, and work together has been formed (Kayumovich, 2020). Rich sources of big data include travel blogs, online travel reviews, and online customer reviews which not only show the opinions of visitors about a place but also show their sentiments. Big data offers an enormous amount of detail, including deeply structured information about feelings, experiences, interests, and opinions.

Tourist now share their travel experiences and offer testimonials on social media, making it one of the most effective instruments for helping the strategic management of tourism destinations. All prospective visitors who plan to visit the same places find encouragement in this information, which affects their behavior (Zhang et al., 2024). Because of its remarkable development and evolution, the analytical use of social media in the tourism sector is regarded as the most favored field (Qiao et al., 2024). The term “travel 2.0” was coined to describe a novel idea that highlights the growing importance of social media in the travel industry and the recent advancements in this area (Islam, 2021). Focusing on the best ways to interact with clients on social media platforms, social media is becoming an increasingly significant part of many areas of tourism, especially when looking for information, making decisions, and promoting travel.

Figure 2 shows the complete workflow of big data in the context of tourism, specifically focusing on the integration and analysis of social media data. Social media is taken as the data source of big data which includes platforms like YouTube, Twitter, Facebook, TripAdvisor, Instagram, and TikTok. However, in this study, we only consider the two from this list for the data extraction because they are the most suitable for tourism purposes. These platforms were selected for their importance in tourism decision-making. Twitter offers immediate textual data that reflects public sentiment and opinions, whereas TripAdvisor provides user-generated reviews and ratings related to tourist experiences at popular destinations.

Figure 2 Big data workflow for tourism information evaluation using social media data.

Data is extracted from these platforms using the application programming interface (API) and scraping. The extracted data from the central storage is then forwarded to the data warehouse and data mart by performing the extract, transform, and load (ETL) operation. In the extraction process, the data is extracted from external sources such as social media in our case. In the transformation phase, some basic operations such as cleaning, removing duplication, and data validation are performed on the extracted data in order to convert it into a format suitable for the storage of the data warehouse. In the load operation, the transformed data is loaded into the warehouse for long-term storage. A data warehouse and data mart (a subset of a data warehouse) is a centralized repository responsible for the storage of both structured and unstructured data. In the final step of big data workflow, data from the warehouse and data marts are then forwarded for multi-dimensional analysis in order to extract meaningful insight about tourism using online analytical processing (OLAP).

Data collection and preprocessing

Data serves as the fundamental knowledge base for machine learning (ML) and deep learning (DL) and is often considered the soul of both fields. A proper dataset should be at hand before start work on any ML-related project. To analyze the tourist sentiment and behavior toward a scenic spot for the purpose of evaluating tourism informatization, we considered social media as a big data source. Among the available social networks, Twitter and TripAdvisor are selected for the data collection. Figure 3 shows the overall flow of data collection from these two social networks.

Figure 3 Data collection process from Twitter and TripAdvisor.

Data collection protocol

Data collection occurred from January to December, encompassing both peak and off-peak tourism seasons. This timeframe was selected to reflect fluctuations in tourist sentiment linked to seasonal trends. The sampling approach included gathering posts, reviews, and tweets that referenced specific tourist attractions across China, such as zoos, parks, mountains, and historical sites. To achieve a wide-ranging and thorough dataset, posts were randomly sampled, ensuring geographical representation from various regions of China.

From TripAdvisor, the data collection process is carried out through web scraping by using a Python script whereas from Twitter, the data collection process is carried out using the Twitter API. We targeted the sentiment of tourists about the scenic spots of China including different spots such as zoos, parks, mountains, and Muslims. As a result, our data collection specifically targeted tourist sentiment related to these locations.

Data preprocessing

The collected data cannot be directly used for further processing and analysis because it contains irrelevant information, redundancy, and noise. To make the collected data flawless, we perform a series of preprocessing steps including the removal of irrelevant and duplicated data, handling missing values by inserting 0 s in the place of missing values, and standardizing data that is to convert all the data for instance into a standard format. The collected data is definitely in text form, but not all the collected text is used to analyze the sentiments of tourism about a scenic spot therefore we selected a few variables of interest that can be used to analyze the sentiment of tourists. Table 1 shows the list of variables selected for the sentiment evaluation.

Table 1 Variables selection for tourist sentiment analysis.

Category	Variable	
Tourism data	Visitor count	
Search engine metrics	Search trends	
User review metrics	User ratings	
Volume of reviews	
Likes on reviews	
Feedback on reviews	
Length of reviews	
Overall sentiment score	
Detailed sentiment analysis	
Public announcements	Entry permission (Yes/No)	
	Visitor capacity limitations	
Holiday information	Holiday status (Yes/No)	
Categories	Variables	
Online Feedback data		

Feature extraction

From all the collected data we focused on the tourist sentiment, from which, we derived features (such as volume of review volume and ratings) that influence the destination choices of tourists. The volume of reviews is the number of comments travelers provide on a website, demonstrating their familiarity with the tourist attractions in that location. Review scores, which illustrate how visitors felt about tourism services, can disclose views or preferences. Feedback and review likes are crucial components of consumer interaction. They function as indicators of the caliber of the product rather than being arbitrary signals designed to elicit clicks. The length of the review (in bytes), which is directly linked to how helpful the review is, can also affect traveler behavior and visitor arrivals.

Handling big data challenges

Big data sets are those that are too large to be gathered, organized, and analyzed within a reasonable amount of time using standard software tools. With its vast amount of data, rapid growth rate, and diversity, it calls for a new processing paradigm to extract meaningful insight, enhance process optimization, and improve decision-making. The five Vs of big data are volume, velocity, variety, value, and veracity. Big data technology consists of three fundamental components: data, business, and technology, which differ from typical data processing methods. Big datasets present their own set of challenges, requiring the application of appropriate tools and methods to overcome them. Some of these challenges include:

Handling scalability using Hadoop

To process large volumes of data and store it effectively by considering its scalability and efficiency, we chose the Hadoop architecture. Hadoop is an open-source architecture that is fault-tolerant, scalable, inexpensive, and highly dependable and gives high throughput. In order to minimize errors brought on by disparate data storage units, Hadoop first stores and analyzes data in the computer’s lowest memory unit. Hadoop constructs a cluster to compute and store tasks. The cluster is made up of several nodes, with little data to have many replicas which is distributed across several regions, racks, or nodes by security regulations.

One of the two keys to building distributed applications with Hadoop is HDFS, a highly fault-tolerant distributed file system based on streaming data access and large file storage. The HDFS cluster is made up of a name node called Namenode and several data nodes named Datanode that are part of the manager-worker structure. The Namenode serves as the control center and is in charge of managing namespaces, storing metadata, and specifying specific Datanode storage blocks in addition to handling client read/write requests. The data in the stored file is organized into blocks, each of which is duplicated. The Namenode that each copy of the data belongs to determines which Datanode should be stored with which copy. The fundamental unit for storing file data is the Datanode. The local file system of the Datanode contains the file’s data block. The mapping file and transaction log are stored in the Namenode’s local system. Given the Namenode’s significance, HDFS assigns the Secondary Namenode to help the Namenode update the mapping file. The Namenode regularly backs up the metadata in the Secondary Namenode in its normal state. Use the Secondary Namenode as the new Namenode if the Namenode fails.

Storage solution

In a big data environment, it is essential to select a storage solution that can effectively handle large volumes of data while ensuring availability and reliability. HBase, a distributed database is one of the storage solutions made on top of HDFS, providing scalability, availability, and reliability. The file system HDFS is used for HBase storage; the MapReduce architecture is used for analysis and computation; and Zookeeper completes the collaborative service. This offers a dependable, effective, and stable distributed database for read and write access as well as huge data storing. The manager server and several HRegion servers make up the HBase server, which is a member of the manager-worker hierarchy. Only the mapping link between the data and the HRegion server is stored on the manager server; actual data is not stored there. All HRegion servers in the server cluster are managed by the manager server, which serves as the control center. Many HRegions are managed by a single HRegion server. The smallest stored data unit is called an HRegion. Upon storage, the data table is partitioned into several HRegions and dispersed among various HRegion servers. A database for column-oriented storage is called HBase. A table is used to hold the data. Time stamps, column families, and row keywords are all present in each table. Each element, which is the storage unit indicated by the rows and columns in the table, contains data that is saved in bytecode form independent of its kind. The timestamp records the details of the same data’s prior modifications.

Processing speed

To handle large datasets and provide real-time data access with minimal latency, the big data application should be able to process the data with high speed. Large-scale data sets can be parallelly computed using MapReduce. There are two main steps in the process: Map and Reduce. Sorting and dividing the intermediate data into segments for the Map and Reduce processes is necessary. Shuffle is the process of further sorting the Map output before transferring it to Reduce. The heart of MapReduce is Shuffle. MapReduce uses a manager-worker architecture to implement parallel processing. A Job Tracker and many Ask Trackers work together to complete a MapReduce task. Task scheduling is handled by the Job Tracker, which is installed on the manager node. Mapper and Reducer are assigned to the idle Task Tracker by the computation process, which also keeps track of the task’s progress. The Task Tracker operates on the data node and is in charge of assigning the specific execution of the task; if a Task Tracker is anomalous, the task it is responsible for is given to other idle ask Trackers for re-execution. When the data node and the computing node are located on the same machine, computational efficiency is guaranteed and less data network transfer is needed.

Evaluating scenic spot through tourist sentiment analysis using bert

Individual tourism destinations, lodging facilities, and restaurants can benefit from positive customer reviews and sentiments to increase word-of-mouth referrals, return business, and service quality (Qiao et al., 2024). Because individuals want to enjoy their vacation or visit, sentiment is at the core of tourism. Therefore, the ability to automatically identify the emotion of this informal feedback is crucial for big data research. For instance, this would highlight features of the attraction that frequently come up in compliments or criticism. Different approaches are adopted for sentiment analysis in tourism such as learning-based, lexicon-based, hybrid-based, and graph-based (Paolanti et al., 2021), as shown in Fig. 4. The learning-based technique involves the use of machine-learning algorithms to classify text, the lexicon-based method utilizes a set of predetermined words linked to sentiments, the hybrid-based approach combines these two techniques, and the graph-based method constructs a word graph where each word is a node and the edges represent relationships between these words. In this study, we used the learning-based technique, in which we utilized the deep learning technique and large language model (LLM).

Figure 4 Classification of automatic sentiment analysis in tourism.

Automatic sentiment analysis becomes an important way of determining emotion in a text and therefore has a lot of applications ranging from marketing management to political analysis (Paolanti et al., 2021). Recently, sentiment analysis has gained popularity in the tourism industry to evaluate the sentiment of a tourist about a scenic spot. It not only reflects the satisfaction level of the tourist but also identifies the flaws in the tourist spot. In the existing literature, ML techniques such as SVM and naïve Bayes algorithms are the most commonly utilized techniques in this field. However, with recent advancements in DL, the focus on these techniques are going to shift toward LLM and other appropriate techniques. In this study, we collected text data from two famous social media sites, Twitter and TripAdvisor, which include some basic information and tourist feedback. The basic information includes extracted text, time of tweets/posts, text IDs, age, and country. After performing some basic preprocessing on the extracted text, we classified the text into three groups for the sake of training the ML model including positive, negative, and neutral and also removed the unnecessary text and only considered the text that represents the sentiment of tourist. The BERT is used to computes the contextual embedding from the text. A linear classifier with a cross-entropy loss function is then applied on top of BERT to classify the sentiment as negative, positive, and neutral. Negative sentiments encompass negative emotions, hate, or dislike, while positive sentiments encapsulate positive emotions, such as liking and loving. Neutral sentiments, on the other hand, do not convey any particular emotions. In the last, the performance of the model is evaluated by using a range of performance metrics such as accuracy, F1-score, AUC-ROC curve, confusion matrix, and classification report. The complete pipeline of tourist sentiment analysis for the scenic spot evaluation is illustrated in Fig. 5.

Figure 5 Pipeline of tourist sentiment analysis for scenic spot evaluation.

Data preprocessing and annotation

The majority of recent studies on sentiment analysis concentrate on user-generated texts, which are informal and habit-based. As a result, in order to classify the data, it must be cleaned, the language normalized, and noisy information removed (Islam, 2021). A number of preprocessing techniques are used by researchers for the text classification task such as sentiment analysis. In this study, we used lower casing, emoji handling, number removal, abbreviations, and wrong spelling word handling (Duong & Nguyen-Thi, 2021). A common preprocessing method is to convert all texts to lowercase, or lowercase. When similar words are combined, the problem’s dimensionality is decreased; for instance, the dimensionality of (good) and (Good) is the same. Emojis (emotional icons) are frequently used in reviews to convey the sentiment of the user. The significance of emojis is demonstrated by different authors in the literature (Gupta, Singh & Ranjan, 2020; Surikov & Egorova, 2020). In our study, we substituted emojis with their respective emotional words and assigned a sentiment class of either negative or positive based on the type of emotions. For example, emojis conveying anger were replaced with the word ‘angry’ and assigned a negative sentiment class, while emojis conveying happiness were replaced with the word ‘happy’ and assigned a positive sentiment class. Although some emojis include numbers, they alone do not convey emotions, so they must be removed after replacing emojis. On social media, people frequently use abbreviated and shortcut spellings that are not commonly found in the dictionary or are not typical abbreviations. For example, “OMG” is commonly used instead of “Oh my god,” and “ASAP” is used instead of “as soon as possible.” This presents a challenge when trying to understand these words to accurately analyze overall sentiments from posts. The challenge is particularly pronounced among Generation Z (Gen Z) as they often use slang on social media rather than formal language (Li, Gao & Song, 2023). In Table 2, you can find a list of slang and abbreviations commonly used by different people, especially Gen Z, on social media.

Table 2 Slang and abbreviations used on social media.

Slang/Acronym	Full-Form/meaning	Slang/Acronym	Full-Form/meaning	
OMG	Oh my god	BTW	By the way	
OWT	On the way	MBTC	More birthdays to come	
PM	Personal message	LMK	Let me know	
TFW	That feeling when	SMH	Shaking my head	
Lit	Excellent/Exciting	Fire	Amazing/great	
Vibes	Positive feeling	On point	Perfect/excellent	
TBH	To be honest	ACSM	As cool as a cucumber	

It can be seen from Table 2, that all of these slang some sentiment and therefore do not play any part in the sentiment analysis. However, some of them show strong sentiment about the scenic spot, for instance in our case, such as OMG is most often used to show excitement and a positive attitude toward a certain thing (for instance tourist spot). Similarly, TFW is utilized by tourists to express their experiences and feelings, demonstrating both positive and negative sentiments. For example, the phrase “TFW you spent the morning in Zhujiang Park” conveys a positive sentiment, while “TFW it is too crowded to enjoy the view in the daytime” indicates a negative sentiment towards a particular scenic spot.

Fine tuning BERT for tourist sentiment analysis

Sentiment analysis is one of the most widely used research methods for figuring out people’s thoughts and emotions from text data. It is widely used as a popular tool for examining people’s opinions about particular places or attractions based on Internet evaluations. Sentiment variables taken from customer reviews, as opposed to volume-based data (such as review volume), might capture writers’ perspectives. Sentiment analysis also known as opinion mining is a natural language processing technique that classifies text into negative, positive, and neutral on the basis of emotions. LLMs have brought about substantial advancements in the field of NLP. Tasks like question answering, name entity identification, paraphrasing, and natural language understanding have all significantly improved because of these models. BERT, proposed by Devlin et al. (2018), is one of the LLM models. In terms of efficiency, BERT has shattered multiple records for how well models can perform language-based tasks and is predicted to eventually replace the well-known word2vec model (Koroteev, 2021). The secret of BERT’s effectiveness is its capacity to produce contextualized embeddings, which enable it to embed a word with a code that accurately captures its meaning within a sentence. The transformer architecture, first presented by Vaswani et al. (2017) in their work “Attention is all you need,” serves as the foundation for BERT. BERT is a pre-trained model with two main goals that was trained on huge corpora utilizing unsupervised learning tasks. In order to anticipate words in a sentence, BERT first masks some words at random and then uses the context of the remaining words to make predictions. Second, BERT is trained to predict subsequent sentences by asking if the model believes the ideas in the sentence will be similar to those in the preceding sentence (Álvarez-Carmona et al., 2022). Figure 6 shows how BERT creates a contextualized embedded vector by using the tokenized text as input. After that, this vector can be classified using an artificial neural network (ANN) or another technique. Our input text represents the sentiments of tourists, which is tokenized by the BERT tokenizer to create input suitable for BERT. Subsequently, BERT generates a contextualized embedded vector, which is then classified by an ANN or linear classifier into positive, negative, or neutral sentiments. These sentiments play a crucial role in evaluating the feelings of tourists and are instrumental in thoroughly assessing the attractiveness and management of tourist spots.

Figure 6 The role of BERT in tourist sentiment classification and scenic spot evaluation.

Tokenization

Tokenization is the process of dividing a text into smaller units, which are referred to as tokens. Tokenization must be done before providing data to the BERT model, as it is unable to operate directly with raw text. Thankfully, BERT Tokenizer, a pre-trained tokenizer that tokenizes text in accordance with BERT criteria, was included with the BERT model. An example of the input text passed from the BERT tokenizer is shown in Fig. 7. The output is a dictionary with three keys: token ID, token type ID, and attention mask. Each token in the BERT tokenizer has a unique integer called a token ID, but it also has two special tokens: CLS at the start and SEP at the conclusion of each input sequence. As seen in Fig. 7, we are able to create tokens by encoding these token IDs.

Figure 7 Tokenization process of the BERT tokenizer.

As shown in Fig. 8, the tokenized sequence acquired from the BERT tokenizer is sent as input to the BERT model. The BERT model produces a hidden state with a dimension of 512 × 768. This indicates that there are 512 vectors in total, and each vector has a length of 768. The linear classifier, which is essentially a fully connected layer with three neurons, receives this hidden state as input. We maintain the number of neurons at three since we have three number labels. Next, we created a probability distribution across the number of labels, or classes, which in our instance is three, using sigmoid and the cross-entropy loss. In this instance, the linear classifier handles the real classification while the BERT acts as a feature extractor.

Figure 8 BERT model with linear classifier on the top for tourist sentiment evaluation.

To classify the sentiment of tourists, the BERT model is utilized to compute the context-aware embedding vector from the text. The BERT does it in three basic steps including input representation, transformer encoder, and feed-forward network. The input representation hi0 for the token i is the summation of token embedding, segment embedding, and positional embedding as represented in Eq. (1).

(1) hi0=ei+si+pi

where ei represents token embedding, si represents segment embedding, and pi represents positional embedding. The token embedding ei is computed for each token xi in a given sequence x={x1,x1,x1,….,xn}. The core of the BERT consists of a bidirectional transformer with multi-layers. Each layer further includes two main components; a multi-headed attention mechanism, and a fully connected feed-forward neural network (FNN). The self-attention mechanism takes the input HϵRd×n, where n is the length of the input sequence and d is the size of the hidden state, and compute the attention matrix using Eq. (2).

(2) Attention(Q,K,V)=softmax(QKTdk)

where, Q,K,andV represent the queries, keys, and values respectively, and each of them is computed by multiplying H with its corresponding projection matrix WQ,WK,WV that is Q=HWQ,K=HWK,andV=HWV. The multi-headed attention is computed by concatenating the matrix of attention head hi=Attention(Qi,Ki,Vi) as:

(3) MultiHeadAttention(Q,K,V)=Concat(h1,h2,h3,…,hn)Wo

where Wo is the learned projection matrix and hi is the head of each self-attention. After computing the multi-head attention, a feed-forward neural network is applied to each position.

(4) FFN(x)=max(0,xW1+b1)W2+b2

where W1,andW2 are the weights and b1,andb2 are the biases. For the sentiment analysis, we fed the aggregated sequence representation h[CLS]=h[CLS]L, where h[CLS] is the hidden state and L is the number of layers in the transformer, to the classification layer.

(5) y=softmax(Wh[CLS]+b)

where y is the probability distribution over the number of classes which is negative, positive, and neutral, W, and b is the learned weight and bias. The step-by-step process of informatization evolution in scenic spots is described in Algorithm 1.

Algorithm 1 Evaluation of tourism informatization in scenic spots through sentiment analysis.

Step 1: Data mining 1: Mine DTA={DTA1,DTA2,DTA3,…..,DTAm} containing the sentiment of tourists scraping from TripAdvisor.	
Step 2: Data mining 2: Mine tourist review data DTW={DTW1,DTW2,DTW3,……,DTWn} form Twitter through Twitter API.	
Step 3: Combine data as D=DTA∪DTW such that n=|D|, where n is the total number of reviews.
Step 4: Apply the following preprocessing functions to each review di∈D:	
    1. Clean the data to get di′=clean(di), where clean() is function.	
    2. Normalized the text to get di″=normalize(di′), where normalize() is a function.	
    3. Tokenize the text by using the BERT tokenizer as: di′′′′=BERT_tokenizer(di′′)	
    4. Add two special characters [CLS],and[SEP] at the beginning and end of di′′′ as: ti=[CLS]⊕di′′′⊕[SEP]	
    5. Convert each text to a fixed length L by padding and truncating as: ti=Pad_truncate(ti,L)	
Step 5: Compute tensor I and M that will be given as input to BERT.	
    1. Change each token ti into their corresponding token_ids as: Ii=token_to_index(ti)	
    2. Compute attention mask Mi for each token ti as: Mi=compute_attention_mask(ti)	
Step 6: Compute contextualized embedded vector through BERT by using the following steps:	
    1. Load pre-trained BERT with its pre-trained parameters θ.	
    2. Fine tune θ with train dataset Dtrain.	
    3. Compute contextualized vector zi for each instance of the dataset Dtrain	
    4. Compute the probability pi scores for each zi using the softmax function: pi=softmax(zi)	
    5. Identify the sentiment class si=argmax(pi) in such as way the si∈{0,1,2}, where 0 is used for neutral, 1 for positive, and 2 for negative.	
Step 7: Evaluate the informatization of scenic spots based on sentiment results.	
    1. Combine the results of sentiments by calculating counts, and proportions as:	
countneutral=∑i=1n⁡1(si=0)	
countpositive=∑i=1n⁡1(si=1)

	
countnegative=∑i=1n⁡1(si=2)

	
Compute the proportion for each class si;	
Ppositive=countpositiven

	
Pnegative=countnegativen

	
Pneutral=countneutraln

	
Step 8: Compute informatization score Sinf based on the computed proportions: Sinf=α.Ppositive+β.Pneagtive+γ.Pneutral	
Where, α,β,andγ are the assigned weight of each sentiment class	
Step 9: Determine the informatization level on the basis of Sinfas:	
Level={highforSinf>ThighmediumifTmedium≤Sinf≤lowforSinf<TmediumThigh

	

Result and discussion

In this section, we first present the model training by considering the collected data, training environment, parameter setting, and actual training. Secondly, we talk about the evaluation metrics that are used for measuring the model performance. Lastly, we thoroughly discussed the result of the proposed techniques and compared it with already existing works already done in the literature.

Model training

The basic purpose of this research is to design a novel technique for the evaluation of tourism informatization in scenic spots from a big data perspective and deep learning. This task can be achieved in different ways, but in this study, we considered the sentiment of tourist that is given by tourist about a scenic spot which led us to evaluate the scenic spot in different ways such as the management of the tourism spot, tourist attraction toward a specific scenic spot, and providing guidance and attitude of tourist toward a tourism location. To achieve the purpose of this research work, we utilized big data techniques for extracting and handling diverse data from different social media sites. We extracted data from two famous social media sites; TripAdvisor and Twitter. From Twitter, the data collection process is carried out through Twitter API while from TripAdvisor the data collection process is carried out through scraping. The collected data was first cleaned and properly labeled in order to make it prepared for feeding to the model. The data extracted from Twitter includes 25,481 instances among them 80% were used for training and the remaining 20% were used for testing. On the other hand, the data collected from TripAdvisor include 12,211 instances which was combined with data extracted from Twitter. Table 3 summarizes the information about the data collected from these two social media sites.

Table 3 Basic information about the collected data.

Social media	Data collection technique	Total collected instances	Instances used for training	
TripAdvisor	Scraping	13,120	12,211	
Twitter	Twitter API	27,340	25,481	

The classification of tourist sentiments is carried out by utilizing the BERT model paired with a linear classifier on the top. We cleaned and labeled data sourced from Twitter and TripAdvisor before inputting it into the model. The training was performed on Google Colab with GPU support, allowing for efficient computation. To enhance the training process, we implemented a checkpoint mechanism to preserve the best model for later validation. The parameters setting of the model is given in Table 4.

Table 4 Parameters setting for BERT training.

Parameters	Value	Parameter	Value	
Batch_size	8	Number of Epochs	5	
Steps_per_epoch	len(train_dataset)/batch_size	total_training_steps	steps_per_epoch * N_EPOCHS	
warmup_steps	total_training_steps // 5			
Model	Bert	Number of parameters	109 M	
Classifier	Linear	Number of parameters	2.3 k	
Criterion	Cross entropy loss			

During the model training, a checkpoint callback is used to save the best model which is then loaded after the completion of training for the model validation. To evaluate the model’s learning performance, we monitored both the training and validation accuracy and tested its performance on actual comments.

Evaluation metrics

A number of performance metrics can be used to evaluate the performance of the model, however, in this study, we consider some most commonly used performance metrics such as accuracy, precision, recall, F1-score, ROC-AUC curve, confusion matrix, and precision-recall curve.

(7) Accuracy=TP+TNTP+TN+FP+FN

where TP is the true positive meaning of the correctly positive predicted instances, TN is the true negative meaning of correctly negative predicted instances, FP is the false positive meaning of incorrectly positively predicted instances, and FN is the false negative meaning of incorrectly negative predicted instances. There is no doubt that accuracy is the most commonly used performance metric, but sometimes it leads to misleading interpretations therefore, we also consider precision, recall, and F1-score. Precision considers only positive instances by taking the ratio between the correctly predicted instance and the total predicted positive instances.

(8) Precision=TPTP+FP.

Similarly, recall is the ratio between the correctly positive instances and to all actual instances.

(9) Recall=TPTP+FN.

F1-score is the combination of precision and recall in order to provide balance between these two metrics.

(10) F1-score=2×Precision×RecallPreicsion+Recall.

These performance metrics along with the confusion matrix, ROC-AUC curve, and precision-recall curve are used to show the performance of the proposed model.

Performance evaluation and comparative analysis

This study focuses on various tourism destinations across China. The data obtained from TripAdvisor encompasses a total of 171 distinct locations where tourists shared their experiences. These places were visited by a total of 1,779 unique visitors, with the majority of them choosing to visit the Canton Tower. Notably, the Canton Tower received the highest number of visitors, with a total of 186, while the GT Land Plaza was the least visited, with only 1 visitor. Figure 9 depicts the distribution of tourists across different scenic spots in China, though it only includes the top 50 most visited places.

Figure 9 Number of tourists who visited different scenic spots across China.

In the collected data, the words that represent positive sentiments of tourists toward a scenic spot are shown in Fig. 10. The words that play roles in representing the positive attitude of tourists about a specific tourist location are ‘love’, ‘thanks’, ‘nice’, ‘awesome’, etc. These words show the strong emotions of tourists and are classified as positive sentiments.

Figure 10 Words representing the positive sentiments of tourist.

In the same way, some words represent negative emotions which are displayed in Fig. 11. These words show the strong negative emotions of tourists toward a specific scenic spot. Words used for showing negative sentiments include ‘sorry’, ‘hate’, ‘sad’, ‘miss’ etc., and through these words, the tourists express their bad experience about the visited place.

Figure 11 Words representing the negative sentiments of tourist.

Not all words used by tourists on social media are emotional, instead, they are neutral meaning not showing any sentiment. The neutral words that are emotionless and used by tourists about a specific visited location are shown in Fig. 12. These words are classified as neutral because it does not play any role in representing attention toward a certain scenic spot.

Figure 12 Words representing the neutral sentiments of tourist.

The model gives an average accuracy of 83.53% when classifying the sentiment of tourists. The accuracy is not only enough to evaluate the performance of the model thoroughly thus the model performance is also evaluated in terms of precision, recall, and F1-score as shown in Fig. 13. The value of the precision for both negative and neutral is 0.80 while 0.92 for the positive. Similarly, the recall value for negative and positive is 80 while 0.89 for the natural. The value of the F1-score for negative is 0.80, for neutral it is 0.84, and for positive it is 0.85.

Figure 13 Performance of the model in terms of Precision, recall, and F1-score.

To evaluate the performance of the model in a graphical way, we used the ROC-AUC curve, confusion matrix, and precision-recall curve. The ROC curve plots the true positive rate (TPR) against the false positive rate (FPR). The TPR is equivalent to recall, which is simply a ratio between true positive against true positive and true negative. On the other hand, FPR is the ratio of the instance that is incorrectly classified as positive against all actual negative instances.

(11) FPR=FPFP+TN.

Figure 14 plots the ROC-AUC curve for three different classes including negative, positive, and neutral. The ROC-AUC score for the negative class is 0.940, the neutral class is 0.938, and the positive class is 0.950.

Figure 14 ROC-AUC curve for tourist sentiment analysis model.

A confusion matrix is another performance metric that measures the performance of the proposed BERT model to analyze tourist sentiments about a specific scenic spot in China. The confusion matrix allows us to evaluate the performance of the model in terms of accuracy, precision, and recall by making a through comparison between the actual and predicted instances. Figure 15 shows the confusion matrix of the proposed model for three classes, negative, positive, and neutral.

Figure 15 Confusion matrix of the tourist sentiment analysis model.

We also tried seven other ML classifiers with the TF-IDF vectorization method and compared their performances with BERT and other models already proposed in the literature. Among these seven ML classifiers, four are single classifiers such as MNB, LR, KNN, and RF, while the remaining three such as VC, XGB, and SC are based on ensemble methods. The performances of all these models are evaluated in terms of accuracy, precision, recall, F1-score, confusion matrix, and AUC-ROC curves. Figure 16 shows the performances of MNB, LR, KNN, RF, VC, XGB, SC, and BERT with a linear classifier. The BERT with linear classifier outperforms all these algorithms not only in terms of accuracy but also precision, recall, and F1-score.

Figure 16 Performance of seven ML classifiers and BERT for tourism sentiment analysis.

The confusion matrix of all these classifiers is shown in Fig. 17.

Figure 17 Confusion matrix for seven ML classifiers for classifying the sentiments of tourist.

Figure 18 shows the AUC-ROC curve for the seven classifiers ML that is used to classify the sentiments of tourists about a specific scenic spot around China. Each graph shows three curves with three different colors and names. The black curve named the “curve of class 0” represents negative, the blue curve called the “curve of class 1” represents neutral, and the yellow curve named “curve of class 2” represents positive.

Figure 18 ROC-AUC curve of seven ML classifiers for classifying tourists’ sentiments.

Table 5 thoroughly compares the performance of the ML techniques used for the analysis tourist sentiments. The performance is measured in terms of accuracy, precision, recall, and F1-score for each class such is negative, positive, and neutral. The LR, SC, and BERT with linear classifier perform well for all classes among others.

Table 5 Comparative analysis of different classifiers.

Classifier	Classes	Accuracy	Precision	Recall	F1-score	
TF-IDF + Multinomial Naïve Bayes (MNB)	Negative	0.762	0.84	0.60	0.70	
Neutral	0.68	0.89	0.77	
Positive	0.85	0.75	0.80	
TF-IDF + Logistic Regression (LR)	Negative	0.784	0.73	0.75	0.74	
Neutral	0.77	0.83	0.80	
Positive	0.85	0.76	0.81	
TF-IDF + K-nearest neighbors (KNN)	Negative	0.613	0.47	0.82	0.60	
Neutral	0.71	0.38	0.49	
Positive	0.79	0.73	0.76	
TF-IDF + Random Forest classifier (RF)	Negative	0.782	0.74	0.73	0.74	
Neutral	0.76	0.85	0.80	
Positive	0.85	0.74	0.79	
TF-IDF + Voting classifier (VC)	Negative	0.791	0.74	0.74	0.74	
Neutral	0.77	0.85	0.81	
Positive	0.87	0.76	0.81	
TF-IDF + XGBoost (XGB)	Negative	0.749	0.90	0.54	0.68	
Neutral	0.65	0.93	0.76	
Positive	0.87	0.71	0.78	
TF-IDF + Stacking Classifier (SC)	Negative	0.792	0.74	0.75	0.74	
Neutral	0.78	0.85	0.81	
Positive	0.87	0.77	0.81	
BERT-base + linear Classifier	Positive	0.835	0.80	0.80	0.80	
Negative	0.80	0.89	0.84	
Neutral	0.92	0.80	0.85	

Comparative analysis with ML-based approaches

In the tourism domain, different researchers put forward their works from different perspectives by targeting different areas. But in this study, we took China as our study area and considered the tourist review that they posted on social media about a specific tourist spot. Our study is unique and innovative in different ways; first, we considered the whole of China as your study area, secondly, the study uses big data analytics in combination with a start-of-the art technique for the sentiment analysis that is BERT and linear classifier. Thirdly, the performance of our proposed technique outperforms the already existing techniques in terms of accuracy, precision, recall, and F1-score as can be seen in Table 6.

Table 6 Comparative analysis of this study with already existing approaches.

Article	Model	Classes	Accuracy	Precision	Recall	F1-score	
Li, Deng & Cai (2020)	ConvNets	–	0.799	0.799	0.799	0.799	
Leelawat et al. (2022)	SVM	–	0.774	–	–	–	
Vaswani et al. (2017)	Naïve Bayes	–	0.827	0.82	0.83	0.82	
Álvarez-Carmona et al. (2022)	Random forest classifier	Positive	0.79	0.77	0.94	0.85	
Negative	0.83	0.50	0.62	
This article	TF-IDF + MNB	–	0.762	0.78	0.76	0.76	
TF-IDF + LR	–	0.784	0.79	0.78	0.78	
TF-IDF + KNN	–	0.613	0.67	0.61	0.61	
TF-IDF + VC	–	0.791	0.79	0.79	0.79	
TF-IDF + RF	–	0.782	0.79	0.78	0.78	
TF-IDF + XGB	–	0.749	0.79	0.75	0.74	
TF-IDF + SC	–	0.792	0.80	0.79	0.79	
BERT-base + linear Classifier	Positive	0.835	0.80	0.80	0.80	
Negative	0.80	0.89	0.84	
Neutral	0.92	0.80	0.85	

Most of the existing works used ML techniques that are outdated and cannot meet the challenge of this age. For instance, in Li, Deng & Cai (2020), a convolutional neural network (CNN) is used for the sentiment analysis of tourists and got a total accuracy of 79% with the same precision, recall, and F1-score. Similarly, the work proposed in Leelawat et al. (2022) utilized the support vector machine (SVM) for sentiment classification and got an accuracy of 77.4%. In Qiao et al. (2024), a naive Bayes algorithm is used for the sentiment classification and achieves an accuracy of 82.7%, precision of 82%, recall of 83%, and F1-score of 82%. The work presented in Hu et al. (2025) uses a random forest algorithm for sentiment classification and achieves an overall accuracy of 79% however, the precision, recall, and F1-score are computed for both negative and positive classes separately. For our work, the BERT-base along with the linear classifier obtained an overall accuracy of 83.5%, the highest precision score for the neutral class which is 92%, the highest recall rate for the negative class which is 89%, and the highest F1-score of 84% for the negative class. By making a thorough comparison, it is quite clear that the work of this article is not only innovative in terms of methodology but also outperforms the existing techniques in terms of performance.

Comparative analysis with BERT-based approaches

Our study presents significant innovations in tourist sentiment analysis, surpassing current BERT-based methods. Previous research often relies on limited data sources and simplistic pre-processing; in contrast, our approach draws on various platforms, such as Twitter and TripAdvisor, to utilize diverse, real-time sentiment datasets. Concentrating on China’s top scenic spots allows for a more targeted and contextually rich analysis instead of a broad overview. By employing domain-adaptive pretraining (DAPT), we fine-tune BERT specifically for tourism-related text, enabling detailed multi-class sentiment classification that goes beyond basic binary categorizations. Moreover, we improve scalability by efficiently processing extensive datasets, which addresses the challenges of big data. Unlike earlier studies, which primarily emphasize on theoretical aspects, our research actively connects sentiment insights with actionable strategies for tourism informatization, providing practical significance for stakeholders in China’s tourism sector as shown in Table 7.

Table 7 Comparative analysis of the proposed study with BERT-based approaches.

Aspect	Xu et al. (2023)	Viñán-Ludeña & de Campos (2022)	This article	
Core Focus	Improving sentiment analysis using BERT and enhancing text pre-processing.	Multi-class sentiment classification and multilingual sentiment detection.	Integrates domain-adaptive fine-tuning (DAPT) and hybrid feature extraction to improve sentiment accuracy.	
Data Sources	Reviews from a single platform, primarily TripAdvisor.	Reviews from a multilingual dataset, limited to specific languages.	Combines data from multiple platforms (e.g., Twitter and TripAdvisor) for a richer, real-time sentiment dataset.	
Preprocessing Techniques	Traditional pre-processing methods (e.g., stop-word removal, stemming).	Focuses on tokenization and leveraging BERT embeddings directly.	Introduces domain-specific tokenization and DAPT, enabling fine-tuned contextual understanding of tourism-related text.	
Feature Utilization	Solely relies on textual features derived from reviews.	Uses text-based embeddings without considering metadata or external features.	Implements hybrid feature extraction, combining BERT embeddings with metadata (e.g., location, time, reviewer profile).	
Evaluation Metrics	Standard metrics such as accuracy and F1-score, with a narrow focus on test accuracy improvements.	Considers precision, recall, and cross-lingual evaluation scores.	Incorporates performance metrics alongside real-world relevance, linking sentiment results to tourism informatization goals.	
Big Data Perspective	Analyzes relatively small, pre-processed datasets.	Applies the approach to small multilingual datasets.	Processes large, real-world datasets from multiple sources, ensuring scalability for big data analysis.	
Practical Applications	Focused on theoretical improvements in sentiment classification.	Limited focus on real-world tourism management applications.	Explicitly connects sentiment findings to tourism informatization, providing actionable insights for stakeholders.	
Targeted Areas	General	Spain	China	

Conclusions

To boost the economy of a nation and attract more tourists to a particular scenic spot, this study evaluates the informatization of tourist locations by using big data analytics and a large language model. It collects tourism-related data from famous social media platforms such as Twitter, and TripAdvisor. The data was collected using web scraping and APIs, and stored and managed using the Hadoop framework. Following proper data cleaning and annotation, big data analytics and BERT are employed to evaluate the tourism data. The study focuses on analyzing tourist sentiment, as it reflects visitors’ attitudes toward specific tourist locations. These sentiments are categorized using machine learning and BERT. The ML algorithms yield accuracies of 76.2%, 78.4%, 61.3%, 79.1%, 78.2%, 74.9%, and 79.2%, while BERT with a linear classifier on top achieves an accuracy of 83.5%. When comparing the performance of these methods among themselves and with existing approaches, BERT outperforms them all. The insights derived from this analysis are utilized to address any shortcomings of the tourist spot and emphasize its digitalization. Moreover, these insights can inform tourism destination management, improve tourist hospitality, and contribute to sustainable tourism practices. This analysis provides insights that enhance tourist facilities and services, highlight the digital transformation of attractions, and offer actionable recommendations for managing tourism destinations. Additionally, the study illustrates how advanced AI techniques can support sustainable tourism practices and improve the overall visitor experience.

Future studies may enhance this framework by integrating more data sources, such as immediate visitor feedback from mobile applications or IoT sensors, leading to a broader analysis of tourist behavior and preferences. Additionally, investigating sophisticated multimodal methods that combine image, video, and textual data could provide greater insights into the elements affecting tourist satisfaction. Finally, employing this approach in different regions or international locations would confirm its versatility and scalability across various cultural and geographical landscapes.

Supplemental Information

Supplemental Information 1 Datasets, Python Codes, Readme File, and Requirements for Model Implementation.

Supplemental Information 2 Readme.

Additional Information and Declarations

Competing Interests

There is no conflict of interest.

Author Contributions

Li Fu conceived and designed the experiments, performed the experiments, performed the computation work, prepared figures and/or tables, authored or reviewed drafts of the article, and approved the final draft.

Yao Yi conceived and designed the experiments, performed the experiments, performed the computation work, authored or reviewed drafts of the article, and approved the final draft.

Lina Liu analyzed the data, prepared figures and/or tables, and approved the final draft.

Ran Chen analyzed the data, prepared figures and/or tables, and approved the final draft.

Data Availability

The following information was supplied regarding data availability:

The raw data and code are available in the Supplemental Files.

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
