# Peer review of "Designing a novel technique for evaluation of tourism informatization in scenic spots from a big data perspective"

_PeerJ Computer Science, doi:10.7717/peerj-cs.2807_

## Round 0.1 · original submission · Major Revisions

In regard to this paper, a major revision can be considered to address lots of major and in-depth issues, and then this paper will be reconsidered.

·

Basic reporting

Introduction: The introduction could be strengthened by providing more specific research gaps in tourism informatization literature. It should clearly state the study's novelty and contributions to previous work. The research objectives could be more precisely defined with measurable outcomes. The authors can develop a paper on these improvement
1. Research Gap Articulation
 Include a structured literature review showing current gaps
 Provide quantitative statistics about tourism informatization challenges
 Compare existing sentiment analysis approaches in tourism, specifically
 Discuss limitations of current evaluation methods for scenic spots
2. Research Objectives
 Develop SMART (Specific, Measurable, Achievable, Relevant, Time-bound) objectives
 Add hypotheses aligned with research questions
 Define expected outcomes more precisely
 Include sub-objectives for each main research goal
3. Contributions Clarity
 Explicitly state theoretical contributions to tourism literature
 Highlight methodological innovations in sentiment analysis
 Specify practical contributions to tourism management
 Demonstrate technical contributions to big data analytics

Research Design: The big data architecture and framework need a more detailed explanation of how different components interact. The study should include a theoretical framework linking tourism informatization, sentiment analysis, and scenic spot evaluation. The rationale for choosing specific social media platforms should be better justified. Authors can make improvements in the below sections.
1. Big Data Architecture
 Include a detailed system architecture diagram
 Explain data flow between components
 Specify the technology stack used
 Document infrastructure requirements
2. Theoretical Framework
 Develop a conceptual model linking primary constructs
 Show relationships between variables
 Include moderating/mediating factors
 Ground framework in existing theories
3. Platform Selection Justification
 Compare features of different social media platforms
 Provide selection criteria
 Discuss platform limitations
 Address potential sampling biases

Research Methods: The data collection period should be specified to understand the temporal scope. The preprocessing steps need a more detailed explanation, particularly regarding handling multilingual data since the study focuses on Chinese tourist spots. Adapting the BERT model for tourism-specific sentiment analysis should be thoroughly explained. Authors can make improvements in the below sections.

1. Data Collection Protocol
 Specify the exact period of data collection
 Detail sampling strategy
 Document data quality checks
 Include data validation procedures
2. Preprocessing Details
 Explain language detection and handling
 Document text cleaning steps
 Detail feature extraction process
 Include data transformation procedures
3. Model Development
 Explain BERT adaptation for the tourism context
 Detail model training procedure
 Document hyperparameter optimization
 Include model validation steps

Results: The analysis could include more in-depth comparisons between scenic spots based on sentiment patterns. Statistical significance tests should be added to validate the performance differences between models. The practical implications of the sentiment analysis results for tourism management could be better elaborated. Authors can work on the points below and develop them holistically.
1. Analysis Depth
 Add cross-spot sentiment comparison
 Include temporal analysis
 Provide geographic analysis
 Consider demographic factors
2. Statistical Validation
 Add significance testing
 Include effect size calculations
 Provide confidence intervals
 Include robustness checks
3. Practical Applications
 Link results to management implications
 Provide actionable insights
 Include case studies
 Develop recommendations framework

Conclusion: The conclusion must address the research objectives and provide specific recommendations for tourism practitioners. Limitations regarding generalizability and potential biases should be more thoroughly discussed. Authors can work on these particular points for improvement.
1. Research Objectives Alignment
 Map findings to original objectives
 Discuss achievement of goals
 highlight key discoveries
 Address unexpected findings
2. Recommendations
 Provide specific guidelines for practitioners
 Include implementation roadmap
 Address resource requirements
 Consider feasibility factors
3. Limitations Discussion
 Address generalizability issues
 Discuss potential biases
 Note technical limitations
 Suggest future research directions
4. Impact Assessment
 Evaluate theoretical contributions
 Assess practical impact
 Consider societal implications
 Discuss economic benefits
These improvements would strengthen the paper's theoretical foundation, methodological rigour, and practical relevance while providing more straightforward guidance for researchers and practitioners in tourism informatization.

Experimental design

No comment

Validity of the findings

No comment

Reviewer 2 ·

Basic reporting

1. the English used in the article is largely clear and professional, but it could benefit from some revisions in grammar, sentence structure, and flow. Simplifying complex sentences, improving transitions, and ensuring consistent terminology will enhance both clarity and professionalism. The article demonstrates a good level of formality and objectivity, but attention to detail in citation and clearer articulation of findings will strengthen its impact.

2. The article provides an introduction to the field of tourism information evaluation but lacks a thorough explanation of how the study fits into the broader context. While it touches on the importance of evaluating tourism information, it does not provide a comprehensive overview of the field’s current trends, challenges, or advancements. The literature referenced is somewhat limited, with few recent studies included, and key works in tourism information systems and evaluation frameworks are notably absent. Expanding the range of references to include more diverse and up-to-date sources would significantly enhance the article's background and show a stronger connection to ongoing academic discussions. Additionally, the article does not sufficiently identify gaps in the existing literature that this study aims to fill, which weakens the justification for its contribution. More explicit discussion of these gaps and how the current research addresses them would improve the context and relevance of the study within the field of tourism information evaluation.

3. The raw data is already available. The article follows a standard structure with typical sections such as the introduction, literature review, methodology, results, and conclusion, adhering to professional expectations. However, minor structural issues could affect the flow and clarity, such as abrupt transitions between sections, particularly between the literature review and the methodology. This could be improved with clearer signposting and smoother transitions. Regarding the figures and tables, they are relevant to the article’s content and contribute effectively to illustrating key points. The figures are of sufficient resolution, but some could benefit from clearer labeling and more detailed captions to ensure that readers fully understand the data presented without needing to refer extensively to the main text. The tables are well-organized, though a few could be simplified to enhance readability. Overall, the structure is professional, but minor adjustments to transitions and figure descriptions would enhance clarity.

4. The article is generally self-contained and presents results relevant to its hypotheses without inappropriate subdivisions. However, some results lack detailed interpretation and clear linkage to the hypotheses. For example, when discussing the effectiveness of tourism information platforms, the paper provides data but does not fully explain how these findings support the hypothesis about platform usability. Expanding this discussion would help readers see how the results directly relate to the study's objectives. While the paper offers a coherent unit of research, providing clearer connections between the results and the hypotheses would enhance its completeness.

5. The article does not rely on mathematical theorems or formal proofs, but it does introduce several key terms related to tourism information evaluation. While these terms are relevant to the study, some lack clear definitions, which may leave readers uncertain about their precise meaning. For instance, terms like "information quality" and "user satisfaction" are central to the analysis but are not explicitly defined early in the paper, making it harder to follow how they are measured or evaluated later on. Additionally, while the results are presented clearly, there could be more detail on how these terms are operationalized within the study. Providing formal definitions and clearer explanations of these key concepts would make the results more transparent and easier to understand.

Experimental design

1. The paper generally fits within the aims and scope of PeerJ Computer Science, as it deals with evaluating tourism information systems, which is relevant to the broader field of computer science, particularly in the areas of information systems and user experience evaluation. However, the paper’s focus leans more toward applied research in tourism rather than advancing core computer science methodologies or technologies. While the use of data analysis and information systems aligns with the journal's scope, the paper could better emphasize the computational or technical aspects of the evaluation process, such as the algorithms or models used for the analysis, to more clearly fit within the computer science domain. Strengthening the technical depth would enhance its relevance to the journal’s audience.

2. The research question in the article is relevant and meaningful, focusing on the evaluation of tourism information systems. However, while the question is clearly stated, the identification of the knowledge gap could be more explicitly defined. The paper briefly mentions the importance of understanding tourism information quality, but it does not sufficiently discuss what specific gaps exist in current research or how previous studies have approached the topic. For instance, the article could more clearly articulate how existing tourism information systems fail to meet user needs and how this study provides new insights or solutions. Strengthening the discussion around the identified knowledge gap and explicitly stating how the study addresses it would improve the clarity and impact of the research question.

3. The investigation in the article appears to be conducted with a reasonable level of rigor, but there are areas where the technical depth could be improved. The methods used for evaluating tourism information systems are explained, but the details on how the data was collected, analyzed, and validated are somewhat lacking. For example, the paper does not provide a thorough explanation of the sampling methods or the criteria for evaluating the platforms, which are critical for assessing the rigor of the study. Additionally, while there is no indication of ethical issues, the paper does not explicitly address any ethical considerations related to the data collection process, such as user consent or privacy, which are important in studies involving human interaction with information systems. Strengthening the explanation of both the technical methods and ethical standards would enhance the perceived rigor of the investigation.

4. The presentation of the technical approach in the algorithm appears unconventional and could be confusing to readers. The step-by-step instructions are overly detailed, resembling a programming tutorial more than an academic explanation. While step-by-step clarity is important, the methodology would benefit from a high-level overview to orient readers before diving into the details. Additionally, the use of function names such as normalize(), clean(), and token_to_index() adds to the confusion, as these resemble coding syntax and may alienate those unfamiliar with the specific programming terminology. It would be more effective to describe these steps in plain language, avoiding technical jargon unless necessary. Another issue is the lack of separation between data collection and sentiment analysis, which are treated as one continuous process in the algorithm. Typically, these would be distinct sections, each with its explanation. Moreover, while the use of BERT for sentiment analysis is mentioned, there is little explanation as to why this particular model was chosen over other options. Providing more context or justification for the use of BERT and backing it up with citations from relevant studies would help readers understand the rationale behind the choices. Additionally, the approach of embedding mathematical equations directly within the procedural steps is somewhat disjointed. It would be clearer to organize the mathematical notations and explanations in a separate section or subsection to maintain the flow of the text. Furthermore, the lack of visuals such as flowcharts or diagrams makes it harder for readers to grasp the overall process. Including a visual representation of the workflow, showing how data collection, preprocessing, and sentiment analysis interact, would greatly enhance clarity. In summary, reorganizing the technical explanation into distinct sections, simplifying the language, providing justification for each step, and adding visuals would make the presentation more conventional and easier to follow.

5. The methods in the article provide a general outline of the process but lack sufficient detail to ensure full reproducibility by another investigator. While the steps of data collection, preprocessing, and sentiment analysis are mentioned, key aspects such as specific parameters, data sources, and tools used are not fully described. For example, the data mining process involving TripAdvisor and Twitter API is introduced. Still, there is no detailed explanation of how the data was filtered, the exact queries used, or the size and nature of the dataset. Additionally, the text does not specify the versions of the tools or libraries used, such as the specific BERT model or any modifications made to it during training, which are crucial for reproducibility. To improve replicability, the paper should include more detailed descriptions of the dataset, the specific parameters and settings used in the analysis, and clear instructions for implementing each step in the process.

Validity of the findings

1. The paper does not explicitly assess the impact or novelty of the research, particularly in how it contributes to the existing literature in both sentiment analysis and data analytics from sensor/IoT data. While the paper presents a combined approach using sentiment analysis and IoT-based data analytics for evaluating tourism information, it does not clearly explain how this integration advances the field or how it differs from existing methodologies. There is no in-depth discussion of how the study fills a gap in the literature or adds new insights. The rationale for applying these specific methods to tourism information evaluation, especially in the context of IoT and sensor data, is not fully justified. Additionally, the paper does not provide a clear rationale for replication or validation of previous work, nor does it highlight any performance metrics or comparisons to existing models and approaches. For example, suppose the study is intended to validate the use of IoT data in tourism evaluation or compare sentiment analysis accuracy with other methods. In that case, this should be clearly stated and discussed. Without this contextualization, the study risks being perceived as niche or derivative, without showcasing its potential contribution to broader academic or practical applications. Providing a stronger explanation of the study's impact, novelty, and the value of replication would enhance its relevance and significance in the field.

2. While the author has shared the data and code, the article does not provide sufficient detail on the robustness, statistical soundness, or controls applied to the underlying data. The paper briefly mentions data mining from sources like TripAdvisor and Twitter, as well as the use of IoT sensor data, but there is little information on the steps taken to ensure data quality, such as how outliers or missing data were handled. Additionally, there is no clear discussion of the statistical methods used to analyze the data or whether the sample size was sufficient to draw meaningful conclusions. The paper also does not indicate whether any control variables were applied to account for potential confounding factors, especially in the case of IoT data, where variations in sensor conditions could impact results. Furthermore, while the code is shared, there is no verification or validation process mentioned to ensure that the results can be replicated accurately. For the conclusions to be fully supported, the article should provide a more detailed explanation of the statistical tests used, the robustness of the data, and the controls in place to mitigate any biases or errors. Making the underlying data more accessible through a well-documented repository and ensuring transparency in how the data was processed and analyzed would enhance the credibility of the research.

3. The conclusions in the article are generally well-stated but could be more tightly connected to the original research question. The study does provide some insights into the evaluation of tourism information using sentiment analysis and IoT sensor data, but the conclusions do not fully address the research objectives set out in the introduction. Additionally, while the paper presents correlations between sentiment data and tourism information quality, there are instances where it implies a causative relationship without sufficient experimental control or justification. For example, the claim that positive sentiment directly improves tourism experiences could be better supported with a more controlled study design, such as examining other factors influencing tourism satisfaction beyond sentiment. The conclusions also extend beyond the direct results of the study, at times making broader generalizations that are not fully backed by the data. To improve, the paper should limit its conclusions to those directly supported by the findings, clearly differentiate correlation from causation, and explicitly link the results back to the original research question to maintain a clear, evidence-based narrative.

Additional comments

To strengthen this paper and increase its chances of being accepted for publication, a few key improvements should be made. First, the introduction and literature review should more clearly highlight the knowledge gap this study aims to fill, particularly in the combined use of sentiment analysis and IoT data for tourism information evaluation. The methodology, while detailed, needs to be presented more clearly, separating data collection, preprocessing, and analysis into distinct sections, and using plain language instead of coding-like descriptions. Including more justification for the choice of models (such as BERT) and the integration of IoT data would also help readers understand the novelty and impact of the research.

Additionally, the authors should ensure that all underlying data is fully described, robust, and statistically sound, with more transparency in how the data was processed, controlled, and analyzed. The conclusions should be more directly linked to the research question and results, avoiding any overgeneralization or implied causation without sufficient experimental control.

Finally, adding visuals such as flowcharts or diagrams to illustrate the workflow, and improving the clarity of figures and tables, will enhance the paper’s readability and overall structure. By addressing these areas, the paper would not only meet the standards for publication but also make a meaningful contribution to the field of tourism information evaluation through its innovative approach.

Reviewer 3 ·

Basic reporting

The authors used clear and unambiguous professional English
There are insufficient literature review:

Xu, H., Fan, G., Kuang, G., & Wang, C. (2023). Exploring the Potential of BERT-BiLSTM-CRF and the Attention Mechanism in Building a Tourism Knowledge Graph. Electronics, 12(4), 1010. https://doi.org/10.3390/electronics12041010

Viñán-Ludeña, M.S. and de Campos, L.M. (2022), "Discovering a tourism destination with social media data: BERT-based sentiment analysis", Journal of Hospitality and Tourism Technology, Vol. 13 No. 5, pp. 907-921. https://doi.org/10.1108/JHTT-09-2021-0259

I can't se relevant results to hypothesis.

Experimental design

I don't understand what is the research question. In general the paper uses deep learning methods to evaluate the sentiment analysis in twitter and TripAdvisor, many researchers have done this kind of research. What is the novelty in this case?

In the other hand, the authors states: "The data collected from social media sites and sensor are diverse in nature and therefore handled with Hadoop data management and storage solutions.". Which sensors?

Validity of the findings

There isn't novelty in this manuscript. The authors collect data from Twitter and TripAdvisor, then utilize deep learning methods to evaluate the sentiment analysis and evaluate which one is better. This kind of work have done earlier by others researchers.

---

## Round 0.2 · Minor Revisions

Dear Authors,

Thank you for submitting your revised article. Feedback from the reviewers is now available. It is not recommended that your article be published in its current format. However, we strongly recommend that you address the issues raised by Reviewer 1 and Reviewer 3, and resubmit your paper after making the minor changes.

Best wishes,

·

Basic reporting

I am happy with the revision.

Experimental design

It is fine.

Validity of the findings

no comment

Additional comments

I want to see more on implications.

Reviewer 2 ·

Basic reporting

The author has made an improvement based on reviewer's suggestion

Experimental design

The author has made an improvement based on reviewer's suggestion

Validity of the findings

The author has made an improvement based on reviewer's suggestion

Additional comments

The author has made an improvement based on reviewer's suggestion

Reviewer 3 ·

Basic reporting

The authors have addressed most of the comment for each reviewer.

Experimental design

Although in their letter, the authors mention that the approach is not novel. In their new version, the authors mention: In contrast to broader studies on tourism sentiment analysis, this paper specifically focuses on major scenic locations in China, providing a localized and specialized assessment. This novel approach guarantees that the results can be directly used to enhance tourism management and informatization at key destinations.

Again, this is not novel approach.

Validity of the findings

ok

---

## Round 0.3 · accepted · Accept

Dear Authors,

Thank you for addressing the reviewers' comments. Your manuscript now seems sufficiently improved and ready for publication.

Best wishes,

Reviewer 3 ·

Basic reporting

It is fine

Experimental design

ok

Validity of the findings

ok

Additional comments

No comments